# Cellular Cytotoxicity and Oxidative Potential of Recurrent Molds of the Genus *Aspergillus* Series *Versicolores*

**DOI:** 10.3390/microorganisms10020228

**Published:** 2022-01-20

**Authors:** Antoine Géry, Charlie Lepetit, Natacha Heutte, Virginie Séguin, Julie Bonhomme, David Garon

**Affiliations:** 1ToxEMAC-ABTE, Centre F. Baclesse, Unicaen and Unirouen, Normandie University, 14000 Caen, France; antoine.gery@unicaen.fr (A.G.); charlie.lepetit@gmail.com (C.L.); virginie.seguin@unicaen.fr (V.S.); bonhomme-j@chu-caen.fr (J.B.); 2Service de Microbiologie, Centre Hospitalier Universitaire de Caen, 14000 Caen, France; 3CETAPS, UFR Sciences et Techniques des Activités Physiques et Sportives Unirouen, Normandie University, 76000 Rouen, France; natacha.heutte@univ-rouen.fr

**Keywords:** *Aspergillus* series *Versicolores*, indoor air, oxidative potential, cytotoxicity

## Abstract

Molds are ubiquitous biological pollutants in bioaerosols. Among these molds, the genus *Aspergillus* is found in the majority of indoor air samples, and includes several species with pathogenic and toxigenic properties. *Aspergillus* species in the series *Versicolores* remain little known despite recurrence in bioaerosols. In order to investigate their toxicity, we studied 22 isolates of clinical and environmental origin, corresponding to seven different species of the series *Versicolores*. Spore suspensions and ethyl acetate extracts prepared from fungal isolates were subjected to oxidative potential measurement using the dithiothreitol (DTT) test and cell survival measurement. The DTT tests showed that all species of the series *Versicolores* had an oxidative potential, either by their spores (especially for *Aspergillus jensenii*) or by the extracts (especially from *Aspergillus amoenus*). Measurements of cell survival of A549 and HaCaT cell lines showed that only the spore suspension containing 10^5^ spores/mL of *Aspergillus jensenii* caused a significant decrease in survival after 72 h of exposure. The same tests performed with mixtures of 10^5^ spores/mL showed a potentiation of the cytotoxic effect, with a significant decrease in cell survival for mixtures containing spores of two species (on A549 cells, *p* = 0.05 and HaCaT cells, *p* = 0.001) or three different species (on HaCaT cells, *p* = 0.05). Cell survival assays after 72 h of exposure to the fungal extracts showed that *Aspergillus puulaauensis* extract was the most cytotoxic (IC50 < 25 µg/mL), while *Aspergillus fructus* caused no significant decrease in cell survival.

## 1. Introduction

Air pollution is a complex and dynamic phenomenon involving the exposure of living organisms to harmful airborne substances. The health effects of indoor and outdoor air pollutants are a major public health issue due to the duration of exposure, even to relatively low concentrations of air pollutants [1,2,3]. According to the World Health Organization, three million deaths per year worldwide are due to stroke (36%), ischemic heart disease (36%), lung cancer (14%), chronic obstructive pulmonary disease (8%), or acute lower respiratory disease (6%) caused by exposure to airborne particles [4]. Airborne physical, chemical, and biological particles form a heterogeneous mixture whose composition varies continuously in space and time [2,3,5]. One of the mechanisms explaining part of the health effects of airborne particles is their ability to synthesize or catalyze the formation of reactive oxygen species (ROS) when they reach the lung cells. These molecules then cause oxidative stress and inflammation of the airways. This ability to synthesize and/or catalyze the formation of ROS is determined by the particles’ oxidative potential (OP), which can be easily measured by biochemical and cell-free assays. Although the physical and chemical properties of airborne particles are well known, we lack data on the biological fraction of these particles, called bioaerosols [6,7,8]. 

Bioaerosols are mainly composed of molds, bacteria, viruses, animal and plant cells, cell fragments, and biological toxins [9]. The development of molds is favored in confined and humid indoor environments (which concerns 30 to 50% of homes) [10]. Because people spend approximately 90% of their time indoors, the study of long-term exposure to airborne fungi is essential.

Molds of the genera *Aspergillus*, *Cladosporium,* and *Penicillium* are the most frequently found in bioaerosols [11,12]. Among the molds of the genus *Aspergillus*, the most recurrent species are *Aspergillus fumigatus*, *Aspergillus niger*, *Aspergillus melleus,* and the species belonging to the series *Versicolores* [13,14]. This series currently includes 17 species, the majority of which have been discovered recently as a result of various phylogenetic studies [15,16,17,18]. These ubiquitous species are present in more than 70% of bioaerosols [19,20,21,22,23] and have been isolated from a wide variety of substrates (food, soil, air, sea water, plants, etc.) [21,22,24,25,26,27]. They are known to synthesize different toxic metabolites such as sterigmatocystin (classified as a potential carcinogen (group 2B) by the International Agency for Research on Cancer [28]), cyclopiazonic acid, 5-methoxysterigmatocystin, dihydroxy-sterigmatocystin, nidulotoxin, averufin, versiconol, and versicolorins A, B, and C [29]. They have also been described as being related to certain symptoms experienced by residents of contaminated habitats such as allergic bronchopulmonary aspergillosis or aggravation of asthma, and to be one of the causative agents of Sick-Building Syndrome (SBS): itching and skin irritation, respiratory tract irritation, coughing, nasal congestion, tiredness, headache, nausea, dizziness, and decreased concentration [10,13,30,31]. In rare cases, most often in immunocompromised patients, they can also cause invasive pulmonary aspergillosis [32,33], aspergilloma [34], endophthalmitis [35], vaginitis [36], kerion [37], or onychomycosis [38].

This study aimed to identify hazards of these new species of the series *Versicolores* to which we are daily exposed indoors. Fungal isolates were collected from bioaerosols and clinical samples. Considering the phenotypic polymorphism of the series *Versicolores* species, the isolates were identified by molecular biology. Spore suspensions and extracts were prepared from the isolates before being submitted to a measurement of their oxidative potential and their toxicity on a respiratory cell line (A549) and a cutaneous cell line (HaCaT). The data collected were analyzed to compare the biological activities of seven species in the series *Versicolores,* and to determine if the oxidative potential could be a predictor of cellular toxicity.

## 2. Materials and Methods

### 2.1. Isolates Collection

A total of 22 isolates were used for this study. Isolates (*n* = 21) were recovered from environmental indoor air samples (bioaerosols) (*n* = 10) and from clinical samples (*n* = 11). The reference strain *Aspergillus amoenus* CBS 245.65 was purchased from the Westerdijk Institute (Westerdijk Fungal Biodiversity Institute, Uppsalalaan, NL-UT, The Netherlands).

Bioaerosols were collected in a cancer treatment center (Centre François Baclesse, Caen, France) and mold-damaged homes (Normandy, France) using a cyclonic biocollector (Bertin Technologies, Montigny-le-Bretonneux, France). For each bioaerosol, four 15 mL samples of sterile water containing 0.02% Tween 80 (Sigma-Aldrich—Merck, Darmstadt, Germany) were collected for 10 min at 300 L·min^−1^.The four samples were then pooled, and decimal dilutions were performed [13]. Undiluted collection liquid and each dilution were grown in triplicates on Malt Extract Agar medium supplemented with 0.02% chloramphenicol (Cooper, Melun, France) (MEA+). Plates incubated at 25 °C were checked daily. Fungal colonies were also isolated and purified on MEA+.

The clinical samples originated from respiratory and superficial skin specimens collected at the Caen University Hospital, for which a mycological examination was prescribed. After inoculation on standard fungal media (sabouraud dextrose agar with chloramphenicol and gentamicin (Bio-Rad, Marnes-la-Coquette, France) and/or sabouraud dextrose agar with chloramphenicol, gentamicin and cycloheximide (Bio-Rad, Marnes-la-Coquette, France)), the isolates were identified as described below.

### 2.2. Molecular Identification

Isolates belonging to the genus *Aspergillus* series *Versicolores* are phenotypically similar, which makes identification by culture and microscopy difficult, even from selective media. We therefore used molecular biology to identify the 21 isolates [15,17].

DNA extraction was performed as described in our previous study [20] using a modified protocol of the Nucleospin™ Plant II kit (Macherey-Nagel, Duren, Germany). Fungal material was transferred to a 2 mL microcentrifuge tube with glass beads. The microcentrifuge tube was incubated twice 15 min at 80 °C and then 15 min at −80 °C. It was then placed into a swing-mill with 400 µL of lysis buffer PL1 for 15 min at 20 Hz, and incubated with 10 µL of RNAse and 20 µL of proteinase K (Promega, Madison, WI, USA) at 10 mg/mL at 65 °C for 15 min. Then, 400 µL of chloroform (MP Biomedicals—Thermo Fisher Scientific, Waltham, MA, USA) was added. The clear supernatant was recovered and extracted as described by the supplier. 

DNA was purified using the NucleoSpin gDNA Clean-up kit (Macherey-Nagel, Duren, Germany) following the instructions of the manufacturer. Quantification and quality of the fungal DNA were performed using a NanoDrop 2000 spectrophotometer (Thermo Fisher Scientific, Waltham, MA, USA) [39]. 

Identification by molecular approach was performed by amplification of the beta-tubulin gene (*BenA*) using Bt2a/Bt2b primers (Eurogentec, Seraing, Belgium). The end-point PCR program included: a hold stage at 94 °C for 5 min, a PCR stage (94 °C for 30 s; 55 °C for 45 s; 72 °C during 90 s) repeated for 35 cycles and another hold stage at 72 °C for 5 min. The PCR products were then sequenced by Eurofins Genomics (Eurofins Genomics, Hamburg, Germany). The obtained sequences were compared using BLAST (Basic Local Alignment Search Tool, NCBI) to the reference sequences of the 17 *Aspergillus* species of the series *Versicolores* (accession numbers: JN853946, JN853963, JN853980, EF652264, EF652273, KJ775086, LC000552, JN854007, KU613371, EF652284, JN853979, JN853970, EF652274, EF652302, JN853976, JN854003, EF652266) [15]. Isolate identification was considered reliable only by having a % ID ≥ 99%.

All *Aspergillus* isolates belonging to the series *Versicolores* (*n* = 22) were grown on MEA+ and stored on slant agar at 4 °C and in a cryoprotective agent composed of sterile water and 10% glycerol (Carlo Erba, Val-de-Reuil, France) at −80 °C before any further testing. We also included a reference strain (CBS 245.65) to validate our identification technique by amplification and sequencing of the *BenA* gene.

### 2.3. Preparation of Calibrated Spore Suspensions

Spores suspensions were made from isolates grown for 10 days on MEA+. The surface of the colonies was covered with collection liquid. The resulting crude suspensions were filtered through sterile cotton to remove hyphae and fungal debris and then through sterile polytetrafluoroethylene (PTFE) filters of 5 µm porosity (Sartorius—Thermo Fisher Scientific, Waltham, MA, USA) to remove clusters of spores stuck together. The spores were then counted on a KOVA® Glasstic slide (KOVA—Thermo Fisher Scientific, Waltham, MA, USA) before performing decimal dilutions ranging from 10^5^ to 10 spores/mL, which are the most commonly encountered concentrations in indoor environments [13]. Final concentrations and quality of spore suspensions were confirmed by flow cytometry on a Cytoflex S hemocytometer (Beckman Coulter, Brea, CA, USA). 

### 2.4. Preparation of Fungal Extracts

Fungal extracts were made from isolates grown for 21 days on MEA+. For all isolates, 12 agar plugs were taken and introduced into 5 mL glass tubes. To each of these tubes, 2 mL of ethyl acetate (Sigma-Aldrich—Merck, Darmstadt, Germany) acidified with 1% acetic acid (Sigma-Aldrich—Merck, Darmstadt, Germany) was added. After vortexing each tube for 30 s, the tubes were centrifuged at 1500 rpm for 15 min. The supernatant (1.5 mL) was collected and filtered through a 0.22 µm pore size syringe tip filter (Thermo Fisher Scientific, Waltham, MA, USA) to remove spores. The extracts were then evaporated in a SpeedVac Plus concentrator (Savant—Thermo Fisher Scientific, Waltham, MA, USA) at room temperature. The dry extracts were stored in the dark at room temperature until use. The dry extracts were then solubilized in a mixture of culture medium and dimethyl sulfoxide (DMSO) (Pan Biotech—Dominique Dutscher, Tourgéville, France) (5%). Each extract was then diluted to obtain four concentrations: 250 µg/mL, 125 µg/mL, 50 µg/mL, and 25 µg/mL.

### 2.5. Oxydative Potential Measurement

Measurement of the oxidative potential of spore suspensions and fungal extracts was performed by the dithiothreitol (DTT) assay. This test is a commonly used cell-free method for assessing the oxidative potential of redox-active chemicals in air. The consumption of DTT (in excess) upon contact with the tested suspensions or solutions is monitored and the depletion of DTT is proportional to the concentration of ROS in the reaction mixture. Existing protocols [8,40] were adapted to suit our tests. The reaction mixture consisted of 550 µL of Dulbecco’s Phosphate Buffer Solution (DPBS) (Gibco—Thermo Fisher Scientific, Waltham, MA, USA) incubated in 2 mL microtubes at 37 °C, to which 50 µL of 0.5 mM DTT (Thermo Fisher Scientific, Waltham, MA, USA) and 50 µL of spore suspension or fungal extract were added. Then, 100 µL of 5,5’-dithiobis-(2-nitrobenzoic acid) (DTNB) (Thermo Fisher Scientific, Waltham, MA, USA) was added after 0, 15, and 30 min of contact staining the reaction medium a yellowish color with an optical density (OD) proportional to the amount of DTT remaining. For each incubation time, 150 µL of mixture was transferred in triplicates to a 96-well microplate. OD was then measured at 412 nm by spectrophotometry (BioTek—Agilent Technologies, Santa Clara, CA, USA). For each assay, a blank (DPBS) was performed in triplicates. For the measurement of the oxidative potential of the extracts, a solvent control (DMSO) was added.

### 2.6. Cell Culture

In order to study the activity of spores and fungal extracts on the skin and at the respiratory level, we chose two cell lines: the A549 (adenocarcinomic human alveolar basal epithelial cells) (ATCC® CRM-CCL-185TM, USA) cell line and the HaCaT (aneuploid immortal keratinocytes) (AddexBio T0020001, USA) cell line. Cells were grown in 25 mL flasks at 37 °C in an environment containing 5% CO_2_ with 5 mL of culture medium containing Dulbecco’s Modified Eagle Medium (DMEM) (Gibco—Thermo Fisher Scientific, Waltham, MA, USA) supplemented with 10% Fetal Bovine Serum (FBS) (Pan Biotech—Dominique Dutscher, Tourgéville, France), 1% penicillin–streptomycin 10,000 U/mL (Gibco—Thermo Fisher Scientific, Waltham, MA, USA), and 0.01% gentamicin 10 mg/mL (Sigma-Aldrich—Merck, Darmstadt, Germany). Cells were resuspended by adding 1 mL of trypsin 0.05% (Gibco—Thermo Fisher Scientific, Waltham, MA, USA) in the case of A549 cells or 0.25% (Pan Biotech GmbH, Aidenbach, Germany) in the case of HaCaT cells. Cells were counted using a KOVA™ Glasstic™ slide (KOVA—Thermo Fisher Scientific, Waltham, MA, USA) and then suspended in DMEM at a cell concentration of 75,000 cells/mL. In each well of a 96-well microplate, 200 μL of the cell suspension was introduced. The microplate was placed in the incubator for 24 h. The culture medium was then replaced with the medium containing the spore suspensions or fungal extracts before being incubated again for 24 to 72 h. For each line and for each condition, six replicates were performed. 

After exposure, cells were fixed with 50 μL of 50% cold (4 °C) trichloroacetic acid (Sigma-Aldrich—Merck, Darmstadt, Germany) for one hour. The plate was rinsed five times with running water and then air-dried. Then, 50 μL of 0.4% sulforhodamine B (SRB) (Sigma-Aldrich—Merck, Darmstadt, Germany) in 1% acetic acid was deposited for 30 min at room temperature in each well to stain the proteins. The plate was then washed four times with 1% acetic acid and air-dried. The dye was solubilized with 100 μL of 10 mM tris(hydroxymethyl)aminomethane (TRIS) base buffer pH 10.5 (Sigma-Aldrich—Merck, Darmstadt, Germany) with 10 min of agitation at room temperature. OD was then measured at 570 nm using a microplate reader by subtracting the reference OD read at 655 nm to eliminate interference. The result obtained for each condition was compared to a culture control (DMEM + 10% FBS) consisting of eight replicates for each experimental condition, expressing the result as percentage of cell survival for each condition compared to the control. For the extracts, a culture control with 5% DMSO was performed in eight replicates in order to estimate the toxicity of the fungal extracts independently of the solvent.

### 2.7. Statistical Analyses

Descriptive statistics were calculated to provide information on oxidative potential. Rates of DTT consumption by spore suspensions and fungal extracts were subjected to the Mann–Whitney test. The percentages of cell survival were subjected to the Kruskal–Wallis test. Only results at *p* < 0.05 were considered statistically significant. Statistical analyses were performed using SAS system v.9.4 (SAS Institute Inc. Cary, NC, USA) and XLSTAT v.2021.2.1.1120 (Addinsoft, Paris, France).

## 3. Results

### 3.1. Isolates Identification

Identification of isolates by amplification and sequencing of the *BenA* gene is presented in Table 1. All sequences are presented in Appendix A.

The reference strain CBS 245.65 was well identified as belonging to the species *Aspergillus amoenus*. The other 21 isolates could be identified as belonging to seven different species of the series *Versicolores*: *Aspergillus amoenus* (*n* = 1), *A*. *creber* (*n* = 7), *A. fructus* (*n* = 1), *A. jensenii* (*n* = 3), *A. protuberus* (*n* = 1), *A. puulaauensis* (*n* = 1), and *A. sydowii* (*n* = 7). *Aspergillus creber* and *A. sydowii* were the most represented species (*n* = 7 isolates for each of the two species). *Aspergillus creber* was only found in bioaerosols, while *Aspergillus sydowii* was only found in clinical samples.

### 3.2. Oxidative Potential of Spores

The DTT consumption rate obtained for the blank (natural oxidation in the reaction mixture in open air) was 0.040 nmol/min. We were able to measure an oxidative potential for each of the 22 isolates of the seven species tested as well as a decrease in DTT consumption rate proportional to the concentrations of the spore suspensions. However, we observed a very heterogeneous oxidative potential even between isolates of the same species. Indeed, the lowest concentration causing a significant increase in DTT oxidation compared to blank for *Aspergillus amoenus* was 10^5^ and 10 spores/mL for strain CBS 245.65 and isolate HAB06, respectively. The same observation was made for *Aspergillus creber* isolates recovered only from the environment (10^5^ spores/mL for 08FM2_A49, 10^3^ spores/mL for HAB07 and HOSP150313_5_98, 10^2^ spores/mL for HAB02 and HAB32, and 10 spores/mL for HAB64 and HOSP050413_5_135), and *Aspergillus sydowii* recovered only from clinical samples (10^4^ spores/mL for 8051266672_C3 and 0062415698_C7, 10^3^ spores/mL for 9071870945_C5 and 0062445522_C8, 10^2^ spores/mL for 4040348777_C2, and 10 spores/mL for 0062445523_C9 and 0112723999_C11). For *Aspergillus jensenii*, a species for which we identified both clinical and environmental isolates, the finding was the same: the lowest concentration significantly increasing DTT consumption compared to blank was 10^3^ spores/mL for 9041799386_C4 and 10 spores/mL for 4070377575_C6 and HAB01. Of the seven species used for the DTT test, *Aspergillus protuberus* was the species with the lowest oxidative potential, while *Aspergillus jensenii* was the species with the highest oxidative potential.

### 3.3. Oxidative Potential of Fungal Extracts

Kinetics of DTT consumption by fungal extracts grouped by species are presented in Figure 1. 

Because the majority of the fungal extracts were yellowish, the final concentration chosen for the DTT assays in the reaction mixture was 25 µg/mL, a concentration for which the OD measurement at 412 nm was not different from the blank (DPBS) in the absence of DTT and DTNB. In microtubes containing only DPBS (blank), we measured an average of 23.87 nmol of DTT after 30 min of incubation (DTT consumption rate of 0.038 nmol/min). Our solvent control (DMSO) showed no oxidative potential (no significant difference from the blank (DPBS)). All extracts showed oxidative potential at a concentration of 25 µg/mL with significantly higher DTT consumption (*p* < 0.05) than those measured for the blank and solvent control. Among the fungal extracts (*n* = 22) of the seven species tested, the extracts (*n* = 2) of *Aspergillus amoenus* showed a significantly higher oxidative potential than the other extracts. However, similarly to the spore suspensions, a significant intraspecific variability in DTT consumption was observed. 

### 3.4. Cell Survival after Exposure to Spore Suspensions

Among all concentrations (10^5^,10^4^, 10^3^, 10^2^, and 10 spores/mL) and exposure times tested (24, 48, and 72 h), only *Aspergillus jensenii* spores at a concentration of 10^5^ spores/mL showed a significant decrease in A549 cell survival (*p* < 0.001) for a mean value of 78.47% ± 4.13 cell survival after 72 h of exposure.

In order to evaluate the impact of the presence of spores of different species, we prepared spore suspensions containing spores of the two, three, four, or five most recurrent species with a final concentration of 10^5^ spores/mL, to which our cells were exposed for 72 h, considering the results already obtained. The mixtures were made from the four species found in the environment: *Aspergillus amoenus*, *A. creber*, *A. jensenii,* and *A. protuberus,* plus *Aspergillus sydowii,* as it was the most frequent species in our clinical samples. As shown in Figure 2, all combinations containing spores belonging to two different species showed a significant decrease in cell survival (*p* < 0.05) for the A549 cell line, with the lowest value for the *Aspergillus amoenus*/*A. creber* mixture (92.04% ± 0.93), and the highest value for the *Aspergillus protuberus*/*A. sydowii* mixture (96.21 ± 1.66). HaCaT cells showed a higher sensitivity than A549 cells at equivalent exposure time (Figure 2B). Indeed, all mixtures containing spores of two or three different species showed a significant decrease in cell survival (*p* < 0.001 and *p* < 0.05, respectively). The lowest percentage of cell survival was observed for the *Aspergillus creber*/*A. jensenii* mixture (95.43% ± 0.57), while the highest percentage of cell survival, significantly lower than the negative control, was observed for the *Aspergillus amoenus*/*A. creber*/*A. sydowii* mixture (97.12% ± 1.22). In the absence of 50% or less cell survival for any of these experimental conditions, no IC50 (50% cell survival inhibiting concentration) could be calculated.

### 3.5. Cell Survival after Exposure to Fungal Extracts

The percentages of cell survival for the A549 and HaCaT lines after 72 h of exposure to the fungal extracts are shown in Figure 3. DMEM with 5% DMSO used to solubilize the fungal dry extracts did not show a significant decrease in cell survival on the A549 and HaCaT lines (98.65% and 98.68%, respectively). Cell survival of the A549 and HaCaT lines after exposure to fungal extracts showed a dose effect for all species. An overall comparison of cell survival percentages obtained for all concentrations and extracts, regardless of species, revealed a greater sensitivity of HaCaT cells to fungal extracts than A549 cells (*p* < 0.0001). For both cell lines, among the fungal extracts tested, *Aspergillus puulaauensis* was the most cytotoxic (IC50 < 25.0 µg/mL for both cell lines), followed by *Aspergillus creber* (IC50 = 116.9 µg/mL for cell line A549 and < 50.0 µg/mL for cell line HaCaT) and *Aspergillus jensenii* (IC50 = 148.9 µg/mL and 209.3 µg/mL for A549 and HaCaT cell lines, respectively). In decreasing order of cytotoxicity: *Aspergillus amoenus* and *Aspergillus protuberus* (for which the comparison of their cytotoxic activity showed no significant difference), *Aspergillus sydowii,* and finally *Aspergillus fructus,* for which no significant decrease in cell survival was observed.

## 4. Discussion

### 4.1. Isolates Identification

Amplification and sequencing of the *BenA* gene allowed the identification of the 21 isolates collected as belonging to seven different species of the series *Versicolores*: *Aspergillus amoenus*, *A. creber*, *A. fructus*, *A. jensenii*, *A. protuberus*, *A. puulaauensis,* and *A. sydowii*. 

*Aspergillus creber* was the most common species in the bioaerosols (7 out of 11 isolates), which was consistent with data on fungal diversity in bioaerosols from the USA [17], Croatia [16], Italy [41], and our previous study [20] (France). *Aspergillus amoenus*, *A. jensenii*, and *A. protuberus* have also been found in bioaerosols in other studies [16,20]. However, the specific diversity here was quite low. Indeed, *Aspergillus cvjetkovicii*, *A. fructus*, *A. griseoaurantiacus*, *A. pepii*, *A. tabacinus*, *A. sydowii*, *A. tennesseensis*, or *A. venenatus* have also been isolated from bioaerosols [16,17,26], which was not the case in our study. 

Regarding the species found in clinical samples, our results were only partially consistent with available clinical data [18,42]: *Aspergillus sydowii* seemed to be the most frequent species found in clinical samples, but we did not find *A. creber* or *A. amoenus*, which were the two most frequent species cited after *A. sydowii*.

### 4.2. Oxidative Potential

To our knowledge, this study was the first to provide data on the oxidative potential of species belonging to the series *Versicolores*. Each isolate showed a significant oxidative potential, but for different spore concentrations. The species whose spores showed the highest oxidative potential was *Aspergillus jensenii,* which is considered as one of the most frequently found species in bioaerosols after *A. creber*. DTT consumption rates were between 0.129 and 0.085 nmol/min for suspensions with 10^5^ and 10^2^ spores/mL, respectively, which was lower than those known for *Aspergillus fumigatus* (between 0.422 and 0.185 nmol/min for spore suspensions with 10^5^ and 10^2^ spores/mL, respectively), but closer to those measured for *Aspergillus brasiliensis* (between 0.240 and 0.007 nmol/min for spore suspensions with 10^5^ and 10^2^ spores/mL, respectively) [8]. Due to the lack of studies on this subject, we have no comparative information regarding the intraspecific variability that we observed with our isolates. 

At a concentration of 25 µg/mL, all extracts showed a higher oxidative potential than the blank. We were able to measure an oxidative potential with an average rate of DTT consumption of 0.116 nmol/min for all species of the series *Versicolores* combined. This value was not significantly different from that obtained for spore suspensions at 10^5^ spores/mL (*p* = 0.16). The species for which the extracts showed the highest DTT consumption was *Aspergillus amoenus,* which has previously been found in air, but is more common in food [17,43]. Since this was the first study of the oxidative potential of fungal extracts, we did not have a point of comparison to determine whether or not these values were high compared to extracts of *Aspergilli* from the series *Versicolores* made with other solvents, or to extracts made with the same solvent but for other species.

Thiols (such as DTT) can interact with spores, and are able to modify their structure by giving a giant cell a shape close to that of chlamydospores [44]. This interaction could account for some of the DTT consumption; however, the incubation time required to obtain these giant cell forms is three days, which is much longer than the 30 min incubation time in our study. As mentioned in the introduction and in previous studies, the species of the series *Versicolores* showed a significant phenotypic polymorphism. This polymorphism was found at the microscopic level with the presence of spores of variable colors, and can be smooth, rough, or verrucous; and at the macroscopic level with a variety of textures and synthesis of pigments and exudates, even between isolates belonging to the same species [17,20,45]. 

Pigments and exudates contain secondary metabolites with various activities: antifungal and antibacterial agents (asperversin, averufins, aspergillomarasmine A) [46,47] or cytotoxic agents (versixanthones) [27]. One of the molecules found in the pigments of *Aspergillus* of the series *Versicolores* is melanin, which, although historically known for its antioxidant properties, in fact has a duality of functional activity through eumelanin (photoprotective and antioxidant) and pheomelanin (phototoxic and prooxidant) [48,49,50]. A variable production of pheomelanin and/or eumelanin or other unidentified compounds from one isolate to another could explain this variability in the oxidative potential measured for the extract from isolates belonging to the same species of the series *Versicolores*.

### 4.3. Cell Survival

For the first time, we exposed cells of the A549 and HaCaT lines directly to spore suspensions and total acidified ethyl acetate extracts of series *Versicolores* species.

The cell survival measured after 24, 48, and 72 h of exposure did not show any significant difference with those measured for the negative control, except for the A549 cell line exposed to spore suspensions at 10^5^ spores/mL of the *Aspergillus jensenii* species, considered as the most frequently found species of the series *Versicolores* in bioaerosols after *A. creber* [16,20]. 

The tests performed with spore suspension mixtures aimed at mimicking the interactions between different species of the series *Versicolores* most frequently and simultaneously found in bioaerosols and their impact on cell survival. After 72 h of exposure, all mixtures containing 10^5^ spores/mL of two species in equivalent amounts showed a significant decrease in cell survival of the A549 and HaCaT lines. We could make the same observation for mixtures containing 10^5^ spores/mL of three different species in equivalent amounts for the HaCaT cell line. This indicated that the simultaneous presence of spores of at least two different species generated interactions leading to the stimulation of secondary metabolite biosynthetic pathways [51]. These metabolites, absent when spores of only one species were present in suspension, caused a decrease in cell survival. In other words, spore toxicity was exacerbated by the concomitant presence of spores of other species. An increase in the species richness of airborne *Aspergilli* of the series *Versicolores* is therefore a risk factor that increases their toxicity. The lack of a significant decrease in cell survival with mixtures of spores from more than three different species probably was related to a relative decrease in the number of spores of each species in the mixture. Similarly, cytotoxicity was higher for mixtures containing spores from two species than those from three different species.

Cell survival tests performed on total extracts with acidified ethyl acetate showed that the HaCaT cell line was more sensitive than the A549 cell line, and that the extract made from *Aspergillus puulaauensis* was the most toxic. However, this mold was not the most common species of the series *Versicolores* in the air [20,22,41]. On the other hand, the two other species that showed the highest decrease in cell survival (*Aspergillus creber* and *A. jensenii*) were the most frequent in the air [16,20]. These species are indeed known to synthesize sterigmatocystin, 5-methoxysterigmatocystin, and versicolorin A and B, which are cytotoxic mycotoxins that can be extracted by acidified ethyl acetate [52,53,54].

### 4.4. A Link between Oxidative Potential and Cell Survival?

All spore suspensions showed a significant decrease in the amount of DTT present in the reaction medium, especially for the 10^5^ spores/mL *Aspergillus jensenii* suspensions.

Only the A549 cell line after 72 h of exposure to *Aspergillus jensenii* spore suspensions at a concentration of 10^5^ spores/mL showed a significant decrease in cell survival. This may suggest a possible correlation between these two tests.

However, measurement of the oxidative potential revealed that extracts of *Aspergillus amoenus* consumed DTT most rapidly, while at an equivalent concentration (25 µg/mL), cell survival was not altered. On the contrary, the oxidative potential of *A. puulaauensis* was not significantly different from that measured for the other species but showed the highest cytotoxicity. 

An overall comparison of the data obtained for oxidative potential and cell line survival for extracts at 25 µg/mL clearly showed that there was no relationship between the results of these two tests. This meant that the oxidation generated by the spores and extracts was not sufficient to decrease cell survival, and that the observed toxicity was due to other mechanisms than oxidation alone, such as a direct activation of caspase-3 by sterigmatocystin [55]. 

### 4.5. Are Clinical Isolates More Dangerous Than Environmental Isolates?

In our study, we observed an important intraspecific heterogeneity in terms of toxicity. Statistical tests performed did not show a higher oxidative potential or cytotoxicity for the clinical isolates than for the environmental isolates. However, it is interesting to note that the most represented species among the clinical isolates was *Aspergillus sydowii,* which grows more easily at 37 °C than the other species of the series *Versicolores* [20].

### 4.6. Limitations

In this first study, the measurement of the oxidative potential was only performed with the DTT test, which is a cell-free test. Similarly, the cytotoxicity tests were conducted on two cell lines (one for skin and one for the airways) using SRB staining. Moreover, other reference strains (such as *Aspergillus amoenus* CBS 245.65) should be tested in future studies.

## 5. Conclusions

In conclusion, DTT tests allowed us to highlight that *Aspergillus jensenii* spores and *A. amoenus* extracts had the highest oxidative potentials. Cell survival assays showed a decrease in cell survival only for *Aspergillus jensenii* spores at a concentration of 10^5^ spores/mL after 72 h of exposure of A549 cells and for all fungal extracts, especially for *A. puulaauensis*. We were able to demonstrate a greater sensitivity of HaCaT cells to the fungal extracts than for A549 cells. The cell survival tests performed with the spore mixtures allowed us to observe a potentiation of the toxicity when spores of different species were present concomitantly in suspension. The comparison of the two methods showed that the measurement of an oxidizing potential alone was not predictive of cellular toxicity.

These first data on the toxicity of spores and extracts of *Aspergilli* species of the series *Versicolores* allowed us to affirm that a great intraspecific variability exists in terms of biological activity, and that these species do not all present the same hazard for human health, hence the need to identify them within bioaerosols. Identification and quantification of new metabolites must be undertaken to explain this variability in biological activity. Studies on toxicity of these metabolites are also necessary to contribute to health risk assessment of molds belonging to *Aspergillus* series *Versicolores*.

## Figures and Tables

**Figure 1 microorganisms-10-00228-f001:**
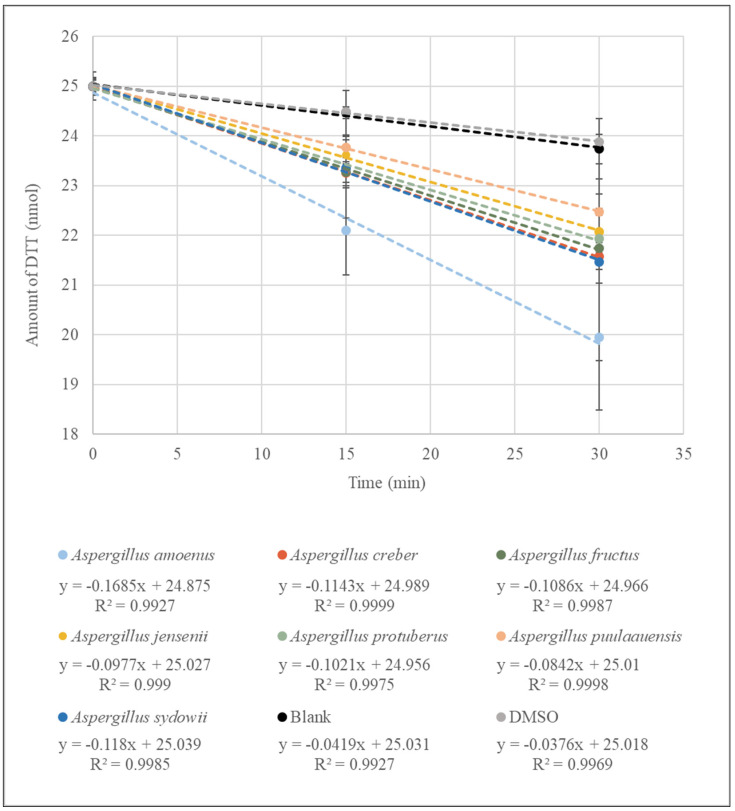
Kinetics of DTT consumption by fungal extracts (*n* = 22) at a concentration of 25 µg/mL of seven species belonging to the series *Versicolores*. Each data point is the average of the measurements obtained in triplicates for isolates of the corresponding species.

**Figure 2 microorganisms-10-00228-f002:**
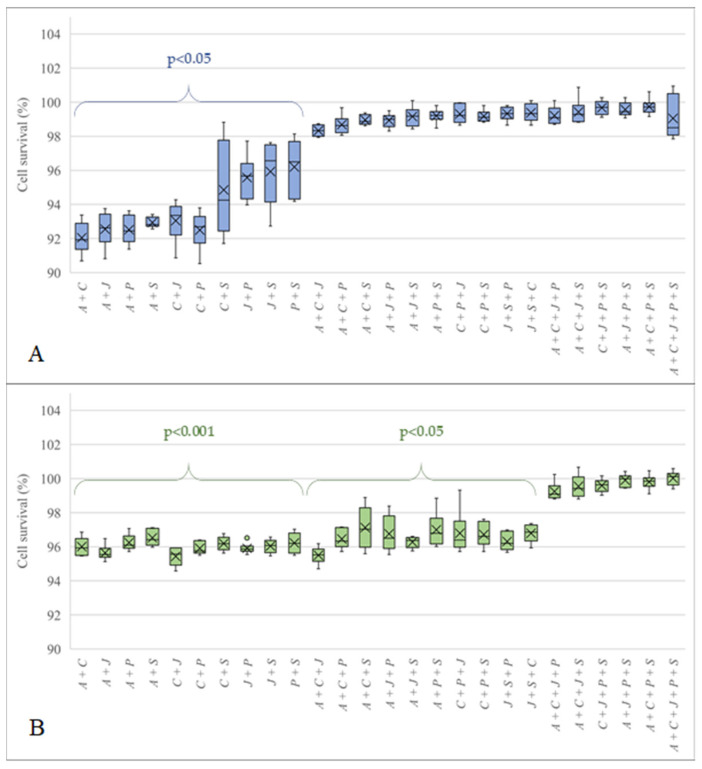
Cell survival of A549 (**A**) and HaCaT (**B**) cells after 72 h exposure to spore suspension mixtures. A: *Aspergillus amoenus*; C: *A. creber*; J: *A. jensenii*; P: *A. protuberus*; S: *A. sydowii*. The *p*-values indicate whether or not the cell survival measured for our assays was significantly different from that measured for the negative controls. Error bars represent the standard deviation calculated with six replicates (when not visible, they are smaller than the symbols).

**Figure 3 microorganisms-10-00228-f003:**
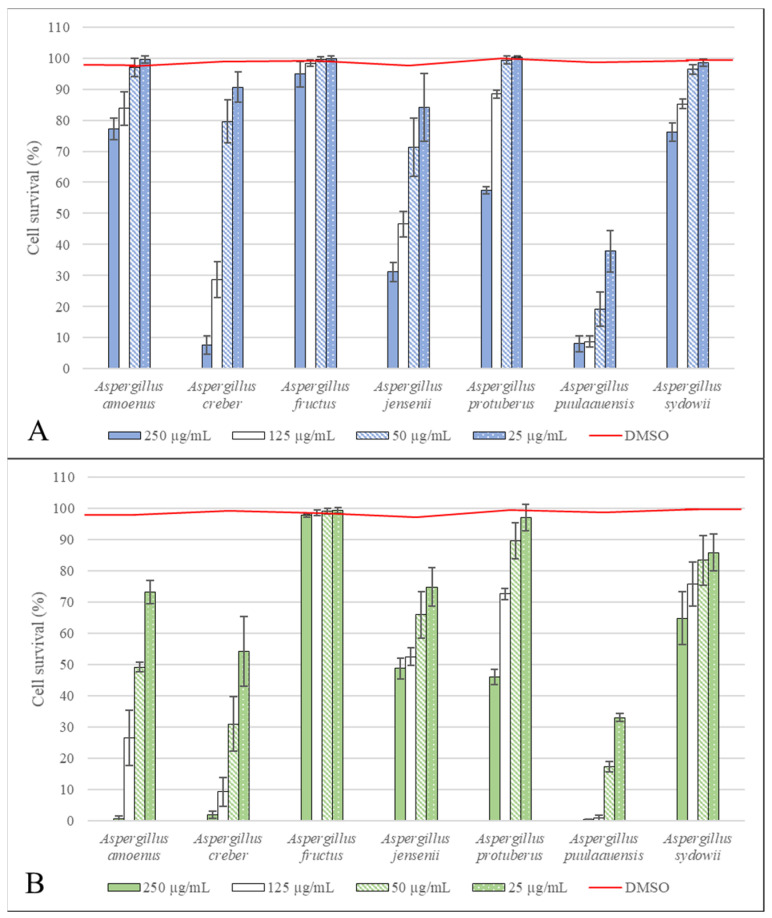
Survival of A549 (**A**) and HaCaT (**B**) cells after 72 h exposure to fungal extracts.

**Table 1 microorganisms-10-00228-t001:** Identification and origins of *Aspergillus* isolates of the series *Versicolores*.

Species	Strain	Origin
*Aspergillus amoenus*	CBS 245.65	Cellophane gas mask
	HAB06	Mold-damaged home
*Aspergillus creber*	HOSP150313_5_98	Cancer treatment center
	HOSP050413_5_135	Cancer treatment center
	08FM2_A49	*Serpula lacrymans*-damaged home
	HAB02	Mold-damaged home
	HAB07	Mold-damaged home
	HAB32	Mold-damaged home
	HAB64	Mold-damaged home
*Aspergillus fructus*	3030204738_C1	Nail of big toe
*Aspergillus jensenii*	HAB01	Mold-damaged home
	9041799386_C4	Scalp
	4070377575_C6	Bronchoalveolar lavage fluid
*Aspergillus protuberus*	HOSP050413_4_129	Cancer treatment center
*Aspergillus puulaauensis*	0102634450_C10	Armpit skin
*Aspergillus sydowii*	4040348777_C2	Bronchoalveolar lavage fluid
	8051266672_C3	Bronchoalveolar lavage fluid
	9071870945_C5	Auditory canal
	0062415698_C7	Bronchoalveolar lavage fluid
	0062445522_C8	Bronchoalveolar lavage fluid
	0062445523_C9	Bronchoalveolar lavage fluid
	0112723999_C11	Bronchoalveolar lavage fluid

## Data Availability

Not applicable.

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
