# Peer review of "Cellular Cytotoxicity and Oxidative Potential of Recurrent Molds of the Genus Aspergillus Series Versicolores"

_microorganisms, 2022, doi:10.3390/microorganisms10020228_

Round 1

Reviewer 1 Report

Results presented in the manuscript of Géry et al. Cellular cytotoxicity and oxidative potential of recurrent molds of the genus Aspergillus series Versicolores are valuable results which contribute to general knowledge of the toxicity of Aspergillus moulds present indoor. The aim of research is well written and explained. The manuscript needs a minor English language editing before being accepted for publication. There are few too long sentences and some sentences are difficult to understand. Material and methods, Results, Discussion and Conclusion sections are very well explained. After suggested language editing, manuscript will be recommended for publication.

Author Response

  • English was improved.
  • The abstract was revised.
  • Long sentences were shorten.
  • Hard to understand sentences were modified.

Reviewer 2 Report

  1. Line 12: instead of … please write etc.
  2. Line 70-75. Authors should also mention allergic bronchopulmonary aspergillosis
  3. Line 89: was purchased to or from?
  4. English needs minor revising
  5. The authors state that the toxicity of spores is exacerbated by the concomitant presence of spores of other species (Line 402-404). Indeed, in Figure 2, combination of 2 or 3 species leads to reduced cell survival in cell lines. However, a combination of 4 or 5 species does not. How can this be explained? This should be noted in the discussion section, since, combination of many species is likely to be more relevant in real conditions in nature.
  6. Since oxidative potential does not clearly correlate with the reduction of cell survival, as the authors note (Line 428), could the authors speculate any potential mechanisms by whom the cell viability is affected in the discussion section?
  7. A limitations subsection in the discussion should be added before the conclusion section. Some limitations include the fact that the study was performed in only one cell line per tissue (one for skin and one for the airways). Alternatively, the authors could perform the same experiments in more cell lines.

Author Response

  • Line 12: instead of … please write etc.

Line 58: sentence was modified.

  • Line 70-75. Authors should also mention allergic bronchopulmonary aspergillosis

Line 63: “allergy” was replaced by “allergic bronchopulmonary aspergillosis”

  • Line 89: was purchased to or from?

Line 83: sentence was modified.

  • English needs minor revising

English was improved and the abstract was revised.

  • The authors state that the toxicity of spores is exacerbated by the concomitant presence of spores of other species (Line 402-404). Indeed, in Figure 2, combination of 2 or 3 species leads to reduced cell survival in cell lines. However, a combination of 4 or 5 species does not. How can this be explained? This should be noted in the discussion section, since, combination of many species is likely to be more relevant in real conditions in nature.

Line 398-402: explanations were added “The lack of a significant decrease in cell survival with mixtures of spores from more than three different species is probably related to a relative decrease in the number of spores of each species in the mixture. Similarly, cytotoxicity was higher for mixtures containing spores from two species than those from three different species.”.

  • Since oxidative potential does not clearly correlate with the reduction of cell survival, as the authors note (Line 428), could the authors speculate any potential mechanisms by whom the cell viability is affected in the discussion section?

Line 426-427: another hypothesis of mechanism was proposed

  • A limitations subsection in the discussion should be added before the conclusion section. Some limitations include the fact that the study was performed in only one cell line per tissue (one for skin and one for the airways). Alternatively, the authors could perform the same experiments in more cell lines.

Line 436-441: a limitations subsection was added

“4.6. Limitations

“In this first study, the measurement of the oxidative potential was only performed with the DTT test which is a cell-free test. Similarly, the cytotoxicity tests were conducted on two cell lines (one for skin and one for the airways) using SRB staining. Moreover, other reference strains (like Aspergillus amoenus CBS 245.65) should be tested in future studies.”.

Round 2

Reviewer 2 Report

The manuscript has been improved during the revision process.